# GenQu: A Hybrid System for Learning Classical Data in Quantum States

## Abstract

Deep neural network-powered artificial intelligence has rapidly changed our daily life with various applications. However, as one of the essential steps of deep neural networks, training a heavily-weighted network requires a tremendous amount of computing resources. Especially in the post Moore's Law era, the limit of semiconductor fabrication technology has restricted the development of learning algorithms to cope with the increasing high intensity training data. Meanwhile, quantum computing has exhibited its significant potential in terms of speeding up the traditionally compute-intensive workloads. For example, Google illustrates quantum supremacy by completing a sampling calculation task in 200 seconds, which is otherwise impracticable on the world's largest supercomputers. To this end, quantum-based learning becomes an area of interest, with the promising of a quantum speedup. In this paper, we propose GenQu, a hybrid and general-purpose quantum framework for learning classical data through quantum states. We evaluate GenQu with real datasets and conduct experiments on both simulations and real quantum computer IBM-Q. Our evaluation demonstrates that, comparing with classical solutions, the proposed models running on GenQu framework achieve similar accuracy with a much smaller number of qubits, while significantly reducing the parameter size by up to 95.86% and converging speedup by 66.67% faster.

## 1 Introduction

In the past decade, machine learning and artificial intelligence powered applications dramatically changed our daily life. Many novel algorithms and models achieve widespread practical successes in a variety of domains such as autonomous cars, healthcare, manufacturing, etc. Despite the wide adoption of ML models, training the machine learning models such as DNNs requires a tremendous amount of computing resources to tune millions of hyper-parameters. Especially in the post Moore's Law era, the limit of semiconductor fabrication technology cannot satisfy the the rapidly increased data volume needed for training, which restricts the development of this field (Thompson et al., 2020).

Encouraged by the recent demonstration of quantum supremacy (Arute et al., 2019), researchers are searching for a transition from the classical learning to the quantum learning, with the promise of providing a quantum speedup over the classical learning. The current state of quantum-based learning inspires alternative architectures to classical learning's sub-fields, such as Deep Learning (DL) or Support Vector Machine (SVM) (Garg & Ramakrishnan, 2020; Beer et al., 2020; Potok et al., 2018; Levine et al., 2019), where the quantum algorithm provides improvements over their classical counterparts. For example, there are quite a number of adoptions of quantum learning algorithms in domains of expectation maximization solving (QEM) (Kerenidis et al., 2019) that speeds up the kernel methods to sub-linear time (Li et al., 2019), Quantum-SVM (Ding et al., 2019), and NLP (Panahi et al., 2019). Employing quantum systems to train deep learning models is rather developed with a multitude of approaches to creating and mimicking aspects of classical deep learning systems (Verdon et al., 2019; Beer et al., 2020; Chen et al., 2020; Kerenidis et al., 2019), with the following challenges: (i), such systems are held back by the low qubit count of current quantum computers. (ii), learning in a quantum computer becomes even more difficult due to the lack of efficient classical-to-quantum data encoding methodology (Zoufal et al., 2019; Cortese & Braje, 2019). (iii),

most of the existing studies are based on purely theoretical analysis or simulations, lacking practical usability on near-term quantum devices (NISQ) (Preskill, 2018).

More importantly, the above challenges would presist even when the number of qubits supported in quantum machines get siginificantly increased: when the number of qubits in the quantum system increases, the computational complexity grows exponentially (Kaye et al., 2007), which quickly leads to tasks that become completely infeasible for simulation and near-term quantum computers. Therefore, discovering the representative power of qubits in quantum based learning system is extremely important, as not only does it allow near-term devices to tackle more complex learning problems, but also it eases the complexity of the quantum state exponentially. However, to tackle the topic of low-qubit counts of current quantum machines is rather sparse: to the best of our knowledge, there is only one paper for the problem of the power of one qubit (Ghobadi et al., 2019). Within this domain, the learning potential of qubits are under-investigated.

In this paper, we propose **GenQu**, a general-purpose quantum-classic hybrid framework for learning classical data in quantum states. We demonstrate the power of qubits in machine learning by approaching the encoding of data onto a single qubit and accomplish tasks that are impossible for comparative data streams on classical machines, which addressing the challenges (i) and (ii). Enabled by **GenQU**, we develop a deep neural network architecture for classification problems with only 2 qubits, and a quantum generative architecture for learning distributions with only 1 qubit, and, additionally, We evaluate **GenQU** with intensive experiments on both IBM-Q real quantum computers and simulators (addressing the challenge (iii)). Our major contributions include:

- We propose, GenQu, a hybrid and general-purpose quantum framework that works with near-term quantum computers and has the potential to fit in various learning models with a very low qubit count.

- Based on GenQu, we propose three different quantum based learning models to demonstrate the potential of learning data in quantum state.

- Through experiments on both simulators and IBM-Q real quantum computers, we show that models in GenQu are able to reduce parameters by up to 95.86% but still achieves similar accuracy in classification with Principal Component Analysis (PCA)(Hoffmann, 2007) MNIST dataset, and converge up to 66.67% faster than traditional neural networks.

## 2 PRELIMINARIES

### 2.1 THE QUANTUM BIT (QUBIT)

Quantum computers operate on a fundamentally different architecture compared to classical computers. Classical computers operate on binary digits (bits), represented by a 1 or a 0. Quantum computers however, operate on quantum bits (qubits). Qubits can represent a 1 or a 0, or can be placed into a probabilistic mixture of both 1 and 0 simultaneously, namely superposition. Superposition is one of the core principles that allows quantum computers to be able to perform certain tasks significantly faster than that of their traditional counterparts. When discussing a quantum framework, we make use of the $\langle bra|$ and $|ket\rangle$ notation, where a $\langle bra|$ indicates a horizontal quantum state vector ($1 \times n$) and $|ket\rangle$ indicates a vertical quantum state vector ($n \times 1$). A qubit, as it is some combination of both a $|1\rangle$ and $|0\rangle$ simultaneously, is described as a linear combination between of $|0\rangle$ and $|1\rangle$. This combination is described in Equation 1.

$$|\Psi\rangle = \alpha|0\rangle + \beta|1\rangle \, , |\Psi\rangle = \begin{bmatrix} \alpha \\ \beta \end{bmatrix} \, , |0\rangle = \begin{bmatrix} 1 \\ 0 \end{bmatrix} \, , |1\rangle = \begin{bmatrix} 0 \\ 1 \end{bmatrix} \tag{1}$$

In Equation 1, the state of $|\Phi\rangle$ describes the probabilistic quantum state of one qubit, respectively $|\phi\rangle$. The values of $\alpha$ and $\beta$ are the probability coefficients and what encode information regarding this qubit's state. Although qubits can exist in both $|1\rangle$ and $|0\rangle$ at the same time, when they are measured for a definite output, they collapse to one of two possible value, where in the case above those values are $|0\rangle$ or $|1\rangle$. The coefficients, $\alpha$ and $\beta$, indicate the square root of the probability that the qubit measures as a $|1\rangle$ or a $|0\rangle$. The

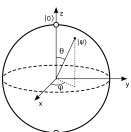

Figure 1: Bloch Sphere

definite states we are measuring the qubit against are based on how
we measure the qubit, measuring as one of two possible measurements. These two possible measurements are two orthogonal eigen-vectors, and can be in any 3-Dimensional direction. This is best visualized and understood by the Bloch Sphere representation of a qubit, as illustrated in Figure 1.

A qubit can be represented by the unit Bloch Sphere visualized in Figure 1. In the case of $|0\rangle$ and $|1\rangle$, we are measuring across the z axis. Although the qubit could be measured against the Y or X axis, once a qubit is measured in a direction and is observed as some vector, the qubit is in that state unless acted upon, therefore making a measurement in Z then X be fraught without further processing. A pure quantum state has data encoded and manipulated through rotations over the Bloch sphere surface. Relating to Equation 1, the $\alpha$ and $\beta$ can be thought of as the states $|\phi\rangle$ distance to the state vectors $|0\rangle$ and $|1\rangle$, where a high $\alpha$ indicates being relatively close to $|0\rangle$ and vice-versa.The power of quantum computing lies in the ability to sample the output repeatedly, thereby providing multiple "answers" for one question.

## 2.2 QUANTUM DATA MANIPULATION

To accomplish data transformation and data encoding, a qubit and its quantum state must be manipulated to encapsulate information onto it. Qubits are manipulated through quantum gates, which in turn manipulates the overall quantum state. These gates can allow for complete manipulation over the Bloch sphere in Figure 1, and more specifically complete manipulation of the quantum state vector, which can describe the state of a mixture of more than 1 qubit. We introduce the few gates that we make use of in this paper in Equations 2 and 3.

$$R_Y(\theta) = \begin{bmatrix} \cos\left(\frac{\theta}{2}\right) & -\sin\left(\frac{\theta}{2}\right) \\ \sin\left(\frac{\theta}{2}\right) & \cos\left(\frac{\theta}{2}\right) \end{bmatrix} R_Z(\theta) = \begin{bmatrix} e^{\frac{-i\theta}{2}} & 0 \\ 0 & e^{\frac{-i\theta}{2}} \end{bmatrix} \tag{2}$$

$$CR_Y(\theta) = \begin{bmatrix} 1 & 0 & 0 & 0 \\ 0 & 1 & 0 & 0 \\ 0 & 0 & \cos\frac{\theta}{2} & -\sin\frac{\theta}{2} \\ 0 & 0 & \sin\frac{\theta}{2} & \cos\frac{\theta}{2} \end{bmatrix} CR_Z(\theta) = \begin{bmatrix} 1 & 0 & 0 & 0 \\ 0 & 1 & 0 & 0 \\ 0 & 0 & e^{\frac{i\theta}{2}} & 0 \\ 0 & 0 & 0 & e^{\frac{i\theta}{2}} \end{bmatrix} \tag{3}$$

The gates above accomplish specific tasks of quantum state manipulation.Equation 2 allows for a single qubit to be manipulated to any position on a Bloch sphere's surface, from any starting point on aforementioned sphere. Equation 3 accomplishes entangling two qubits with controlled rotations. Controlled rotations allow for a single qubits state to be entangled with another. In the case of a controlled rotation gate, a qubits state is manipulated based on whether the control qubit measures as a $|1\rangle$. Although we brush over this for the sake of easier reading, quantum entanglement empowers quantum computers to accomplish phenomenal tasks. These two styles of gates, single qubit rotations and controlled qubit rotations, allow for complete manipulation of quantum states, be it 1 or more qubits.

## 2.3 QUANTUM DEEP LEARNING

Quantum Deep Learning is a relatively new approach to Quantum Machine Learning that takes quantum circuits and applies similar training techniques and learning methods of how classical neural networks work Chen et al. (2020); Garg & Ramakrishnan (2020); Beer et al. (2020).

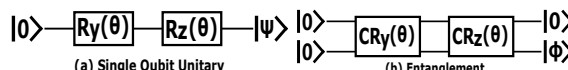

Figure 2: Gate Deep Learning Gate Design

In traditional deep learning layers are often used, where a layer is some large transformation function that takes in a set of inputs, and outputs a set of outputs, where the number of inputs does not necessarily equal the output. These functions are connected in series, sometimes in parallel, and typically trained through the use of back propagation Goodfellow et al. (2016); Chen et al. (2020). This data flow through layers to some output is similar to how quantum circuits operate. Similar to how classical deep learning works, the way this data flows through time is up to the practitioner, who chooses and designs their network according to their needs. Quantum deep learning is approached

through the use of layering gates sequentially. For our paper, our layers are comprised of the gates in Equations 2 and 3. Similar to how deep learning is parameterized by connection weights, these gates are parameterized through rotations ($\theta$). At the end of the quantum circuit, a loss function is described by the practitioner, and the quantum networks parameters $\theta$ are updated iteratively such that the circuits loss is minimized Beer et al. (2020); Crooks (2019).

In the case of binary classification, one can make use of quantum entanglement to pool data down to one qubit channel, which then can be used as the final classification output of the network. These layers are visualized in Figures 3 and 2. In Figure 3 we visualize the grouping of these circuits to be reminiscent of quantum traditional deep learning layers with the oracle approach. Interpreting these operations can

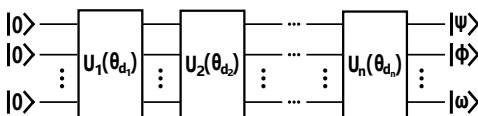

Figure 3: Quantum Deep Learning Layers

be seen as a qubit entering through the left starting in state $|0\rangle$, passing through gates until it has been transformed into state $|\phi\rangle$.

# 3 GENQU FRAMEWORK AND LEARNING MODELS

## 3.1 GENQU FRAMEWORK

Our proposed GenQu framework is illustrated in Figure 4. Before any operation of the framework is performed, the data must be transformed from classical to quantum states. This is done by transforming classical data into applicable quantum rotations, and is described under section 3.2. Following this, the rotations are loaded onto a quantum computer. The quantum circuit preparation section is where the circuit relating to a specific machine learning algorithm is designed. For example, this is where a deep neural network or a convolutional neural

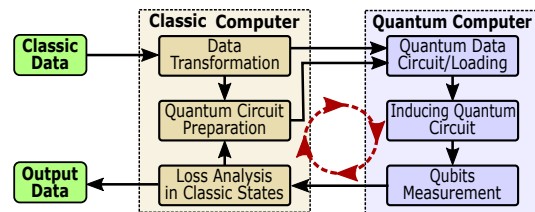

Figure 4: GenQu: A Hybrid Framework

ral networks architecture would be set up, initialized and prepared. This circuit is loaded onto the quantum computer after the quantum data loading section.

Once the circuit is set up, it can be induced. Inducing the quantum circuit results in the quantum state transformation of the input data over the quantum machine learning model. From here, if the output of the model was a quantum state, one could end here and feed it to another quantum algorithm. However, in the case of updating learn-able parameters, the relevant qubits need to be measured. We feed the qubits measurements to a loss analysis section, where we update our parameters accordingly. Once the parameters have been updated, we repeat this process of circuit loading, circuit inducing, and measurement, updating parameters until a desired loss of the network is attained or a predefined number of epochs have run.

## 3.2 DATA QUBITIZATION

Prior to discussing our methods of illustrating the learning power of qubits, we introduce our approach to encoding classical data into quantum states. We encode two dimensions of data per qubit, by the simple two step process outlined in Equations 4 and 5

$$x_1 \xrightarrow[\ |\phi\rangle\ ]{\text{Encoded onto}} = RY(2sin^{-1}(\sqrt{x_1})) \tag{4}$$

$$x_2 \xrightarrow[\ |\phi\rangle\ ]{\text{Encoded onto}} = RZ(2sin^{-1}(\sqrt{x_2})) \tag{5}$$

The value of $x_1$ is encoded along the Z-axis, followed by $x_2$ being encoded along the Y-axis. For these rotations to work, the data along each dimension must be normalized to be in the range of (0,1). This reduction in qubit count is pertinent for the case of quantum machine learning as the state space vector of a quantum system is of tensor rank $2^n$ values, and therefore halving the qubit count provides a $2^{\frac{n}{2}}$ reduction in state space.

### 3.3 SINGLE QUBIT KERNELIZED CLASSIFICATION

When tackling a classification task on a quantum system, we want to encode our data such that the probability of measuring a $|1\rangle$ is comparative with the probability of classifying a data point as $Class = 1$. Therefore, in the case of classical data sets, we can wrap 2 dimensions of data around a qubit such that we maximize the ability of the qubit to classify the data. In the case of the circles data set, a data set comprised of points non-linearly separable, we can wrap the qubit with data points such that the rotation around the Z axis is correlated to the distance from the circle center. This is visualized in Figure 5. This encoding accomplishes on one qubit the encapsulation of 2 dimensions of non-linearly separable data, whilst accomplishing a separation task. For this to be done, two parameters per qubits are used to transfer between Classical data to Quantum state. These parameters are the rotations around the Y axis, proceeded by a rotation around the Z axis.

Translating the classification problem of the data outlined in Figure 5 to GenQu framework, we follow the following approach. We begin by translating our data into rotations according to the functions outlined in 3.2, however the values encoded are vector distances from the circle center. This can be considered both the Quantum Circuit Preparation and Data Transformation components, as the expected measurement under Qubits Measurement is equivalent to the classification of a data point. There is no updating of parameters in this case, therefore we do not iterate and update circuit parameters.

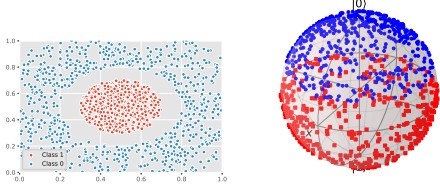

Figure 5: Circles dataset and its qubit representation

### 3.4 QUANTUM DEEP LEARNING ARCHITECTURE

In this paper we make use of the data qubitization techniques outlined above, along side current quantum machine learning techniques to enable highly performent quantum deep learning. Through encoding 2 dimensions of data per qubit, the number of *neurons* in our network input is half the dimensionality of the data set. Quantum deep learning layers comprised of single qubit operations are namely called single qubit unitary layers. In these layers each qubit has a RY and RZ gate appended, thereby adding $2n$ parameters, where n iWs the number of qubits. Another type of layer consists of operations acting on two qubits per gate, where operations are control operations (CRY or CRZ). These are namely Entanglement layers. Entanglement layers entangle all qubits by some learnable amount, performing CRY and CRZ gates on qubits $i$ and $i + 1$ until there are no qubits left to pair. Entanglement layers require $2(n - 1)$ parameters. These gates are visualized in Figure 2. Through the use of the entanglement layer, we can reduce and grow the number of qubits at any time point across a circuit dynamically. In our case of illustrating binary classification, we make use of the entanglement layer to pool down data from the other qubits onto one qubit, which is measured and used as the classification qubit. The probability of the qubit measuring $|1\rangle$ is thought of as the probability of labelling the quantum data that was fed to the circuit as $Class = \mathbf{1}$ similar to how a single output neuron operates of activation function Sigmoid operates in classical neural networks.

Fitting this Quantum Deep Learning model to our GenQu framework, we begin by translating our data into rotations described under 3.2. From here, the practitioner can describe their full quantum deep learning architecture and initialize the parameters. The data is loaded onto a quantum computer in series, with the quantum data loading circuit being appended with the quantum deep learning architecture. The quantum circuit is induced and the classification qubit measured. We feed this result back to a classical computer, calculate our loss and update our parameters accordingly. This is repeated until convergence occurs or sufficient accuracy is attained.

### 3.5 QUANTUM GENERATIVE NATURE

Another powerful use of qubits is in the use of representing data. Through using trainable circuits as discussed above, we can measure two values from one qubit. Therefore, a quantum deep neural network can be trained to mimic some data it is fed by defining some loss function such as the

Figure 6: Quantum Deep Learning Circuit

Mean Squared Error, and generate new samples that are close to what the qubit was trained on. This is similar to Generative Adversarial Networks Goodfellow et al. (2014), however does not take any noise as an input, nor does it require two networks to be used. We do not claim that ours is better, however it is one of the side effects of qubits being used to represent data, and quantum deep learning models. Therefore, we illustrate through the use of quantum deep learning how a quantum deep learning architecture with a tuned loss function can generate data similar to that of the data it was fed, and at a generative diversity significantly greater that is unattainable using similar architectures within its classical counterparts.

Translating a quantum generative state to GenQu framework, we repeat the steps outlined in the Quantum Deep Learning architecture above. However, the only change would be the loss function such that the quantum state instead of a loss function such as cross entropy, could be mean squared error or some other applicable loss function. Furthermore, no data loading for input is necessary, and instead are just loaded as qubits in the state of $|0\rangle$. When generating data, the qubits are measured and sent to the Output Data stream.

## 4 RESULTS

We implement GenQu with IBM Qiskit and Tensorflow Quantum. It is evaluated with the above mentioned three applications, kernelized classification, quantum deep learning and quantum generative nature. We evaluate GenQu on both simulators and IBM-Q quantum computers (mainly Rome). We compare our results with traditional convolutional neural networks with different numbers of parameters. In the rest of the evaluation, we denote CNN - XP to be classical neural networks with x parameters and QNN - XP is quantum based neural networks with x parameters.

### 4.1 THE KERNELIZED CLASSIFICATION

As a proof of a single qubit natural machine learning, we employ the encoding of a circles dataset illustrated in Figure 5 onto one qubit through the radial kernel method. A single qubit has data points encoded as the vector distance from the center of the circle in Figure 5. The qubit is then measured, and the $P(|\phi\rangle) = |0\rangle$ is equivalent to $P(Class = 1)$. Through doing so, we attain $100\%$ accuracy on separating the non-linearly separable data set, whilst maintaining both dimensions of information. Furthermore, when our experiment is run on **IBM-Q's Quantum Computer Rome** $100\%$ accuracy is attained, thereby confirming our model architecture works both on simulators and real quantum computers. This approach, although not novel, is done to illustrate that certain problems can be tackled very efficiently with qubits and how the solution can be successfully run on real quantum computers.

### 4.2 QUANTUM DEEP LEARNING

To evaluate the learning potential of quantum deep learning, and the ability to use fewer data-channels than its classical counterpart, we make use of the MNIST data set. The MNIST data set is an image data set comprised of gray scale hand-drawn digits of resolution 28 by 28. It is infeasible to represent these images on current near-term quantum devices, and hence we make use of PCA (Hoffmann, 2007) to reduce dimensionality from 784 to 4. In this case, we only need to make use of 2 qubits to feed our data to our quantum deep neural network. We provide the deep learning circuit visualized in Figure 6. As can be seen in the circuit, there is a total of 8 parameters. This network is comprised of one single qubit unitary layer (Parameters 0 through 3), one entanglement layer which accomplishes data pooling onto one qubit (Parameters 4 and 5), and finally a single qubit unitary on the final output qubit (Parameters 6 and 7). We compare our architecture to classical deep learning architectures and compare parameter counts when using the same gradient descent

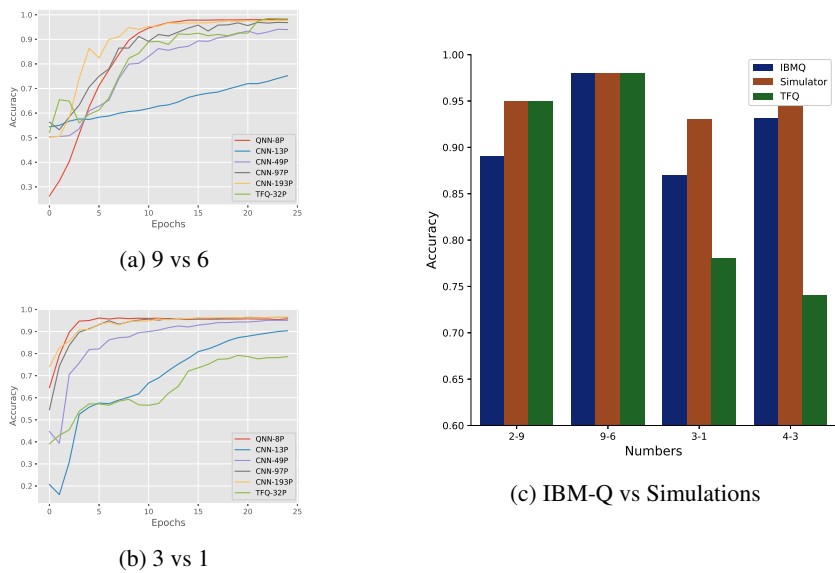

(a) 9 vs 6

(b) 3 vs 1

(c) IBM-Q vs Simulations

Figure 7: QNN (simulation), QNN (IBM-Q Rome), TFQ and CNN Results

approach (Adam Optimizer), same epochs and same data set. The quantum network is trained to perform binary classification of two numbers from the MNIST dataset, where the classification is measured by the Qubit $(0, 1)$ in Figure 6. As for the classical networks, we make use of a network comprised of a middle layer of tensor size $2, 8, 16$ and $32$. The comparative training results are visualized in Figure 7. In Figure 7(a), the numbers 9 and 6 are used to train the data set. The quantum network outperforms all other comparative solutions, with close to equivocal performance of a 193 parameter deep neural network, thereby attaining a 95.86% parameter count reduction, and converging 33.33% faster than said network. However, in the case of 9 and 6, there is a less significant difference between parameter counts than what is observed in other cases, such as 3 and 1. In Figure 7(b), we observe how there is substantial learning ability to be gained from increasing the classical parameter count. However, similarly in this case, the 8 parameter QNN's performance is matched by the 97 parameter CNN, a 91.76% reduction in parameters. We also compare our model to the Tensorflow Quantum (TFQ) MNIST classification example on the same number pairsBroughton et al. (2020). We illustrate that our network outperforms Tensorflow Quantums MNIST classification task in Figures 7 (a) and (b). This illustrates the significant learning potential of quantum networks and specifically the architecture used in this paper. These architectures are able to, in certain cases, reduce parameter counts significantly with no sacrifice to performance. Furthermore, in our case we have encoded two dimensions of data per qubit. Feeding 4 dimensions of data to a deep neural network through 2 *neurons* is impractical, and is a further example of how powerful qubits are in deep learning.

We validate our results by running similar experiments on a real quantum computer using the IBM-Q platform, comparing the accuracy's attained on a simulator to that of a on a quantum computer. These results are visualized in Figure 7(c). As can be seen, for numbers **4-3** and **9-6**, actual quantum computer performance was extremely similar to that of the simulator, with a difference of less than 5%, and 9-6 having a measured difference of 0.2%. However, in the case **3-1** we observe more significant differences between actual Quantum Computing implementation. The largest difference between simulators and actual quantum computers was 7.25% on the **3-1** dataset, which is due to the noise on the quantum computer that depends on the computer itself and the rotated workloads on it (random factors). We further validate our results by comparing them to the TFQ MNIST classification task, and show how in some cases our architecture and network attaining significantly higher accuracy. Specifically, in the case of **4-3** with a 20% improvement and **3-1** with a 22% improvement. However, in certain cases our network attained the same accuracy as TFQ, in the cases of **2-9** and **9-6**.

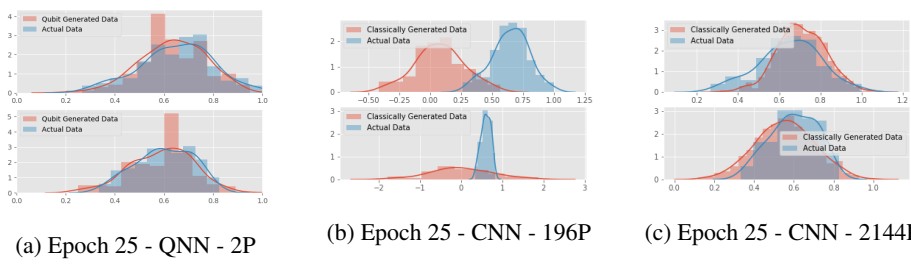

(a) Epoch 25 - QNN - 2P   (b) Epoch 25 - CNN - 196P   (c) Epoch 25 - CNN - 2144P

Figure 8: Single qubit generative model to learn the distribution of PCA MNIST digit 0

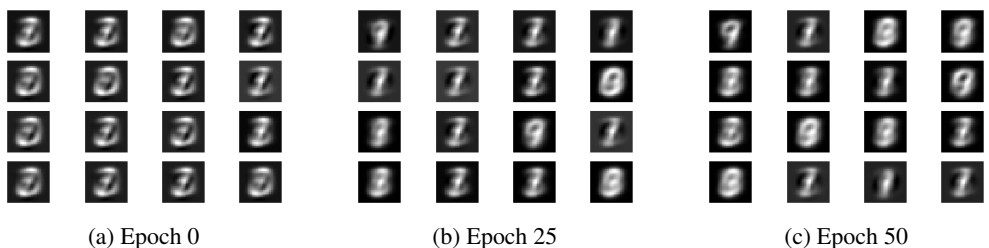

(a) Epoch 0   (b) Epoch 25   (c) Epoch 50

Figure 9: Images Generated by GenQu

### 4.3 QUANTUM GENERATIVE NATURE

Another point of interest is how powerful qubits are in representing data sets. This has significant implications in Generative Adversarial Networks (GANs) and loading data sets. We illustrate this potential by minimizing the distance between a single qubit's quantum state and the MNIST data set PCA'ed to 2 dimensions and of class 0. We illustrate in Figure 8 how a single qubit, visualized by the blue shading, using only 2 parameters (an RY and RZ gate in series on one qubit), can completely mimic the data it was fed (2 dimensions). If sampled, the qubit will generate all samples it was fed as well as generate new unique samples similar to that of which it was fed. We make use of the architecture of a Generative Adversarial Network Goodfellow et al. (2014) to compare this to a classical neural network, and observe how poorly the classical counterpart performs. When given 9800% more parameters (2 vs 196), as visualized in Figure 8(c), the network was still unable to mimic the data fed to the network. The classical network was able to converge when provided with 2144 parameters, as visualized in Figure 8(d). Furthermore, we illustrate the generative potential by sampling from the aforementioned quantum circuits with a shot count of 15, performing reconstructive PCA on the data and plotting the images. As seen in the images the qubit is able to learn from almost nothing in 9(a) to reasonable results after 50 epochs in 9(c). This further goes to illustrate the significant machine learning potential and parameter reduction potential of quantum machine learning.

## 5 CONCLUSION

This paper proposes GenQu, a hybrid and general-purpose quantum framework for learning classical data. It demonstrates the significant expressibility of qubits, and their extensive applications in machine and deep learning.

Based on GenQu, we propose three different learning models that make use of a low-qubit count in near-term quantum computers. In the model for kernelized classification, GenQu is able to to encode the circle dataset onto a single qubit achieve 100% accuracy in the experiment on a real quantum computer. With quantum deep learning model in GenQu, when encoding two dimensions of data per one qubit, it is able to show reductions in parameters equivalent to 95.86%, whilst still attaining a similar accuracy or better than that of classical deep learning models. Finally with respect to qubits learning potential, a single qubit generative model is proposed. It is able to completely learn to generate 2 dimensions of data PCA'ed from the MNIST's 0 class.

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

## A    APPENDIX: EXPERIMENTS ON IBM-Q QUANTUM COMPUTERS

To justify our proposed model, we conducted experiments on multiple IBM-Q sites including Vigo, Ourense, Rome, Bogota and Valencia. Figure 10 is the setup for Kernelized classification on the circle dataset. We ran this circuit with 40 shots (repetition) 20 times on different locations and achieved accuracy of 100% .

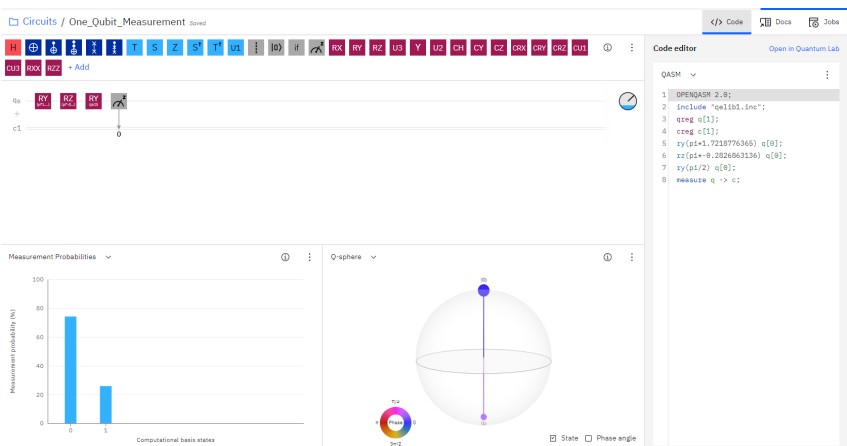

Figure 10: Experiment setup for Kernerlized classification

In PCA MNIST dataset, we train the model on 913 samples of two specific digits (9 and 6) on different IBM-Q locations to justify our founding. Figure 11, 12 and 13 present a sample of each iteration over actual IBM-Q machine which ran for 913 times to get accuracy of 95% on average.

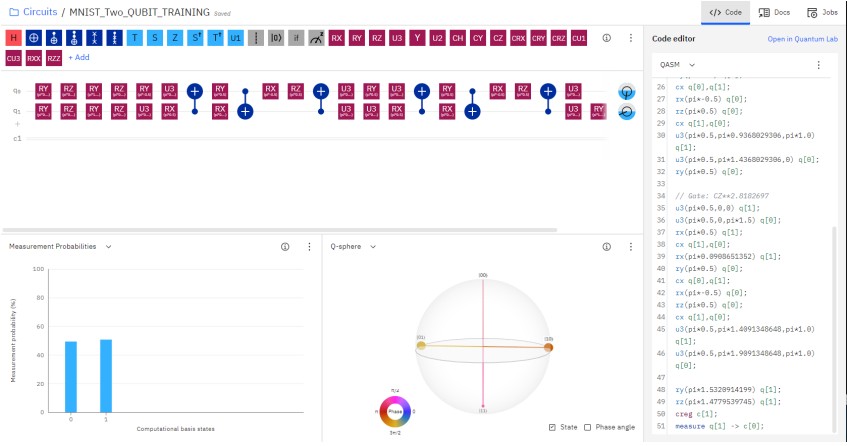

Figure 11: Quantum circuit for binary classification on MNIST dataset (T1)

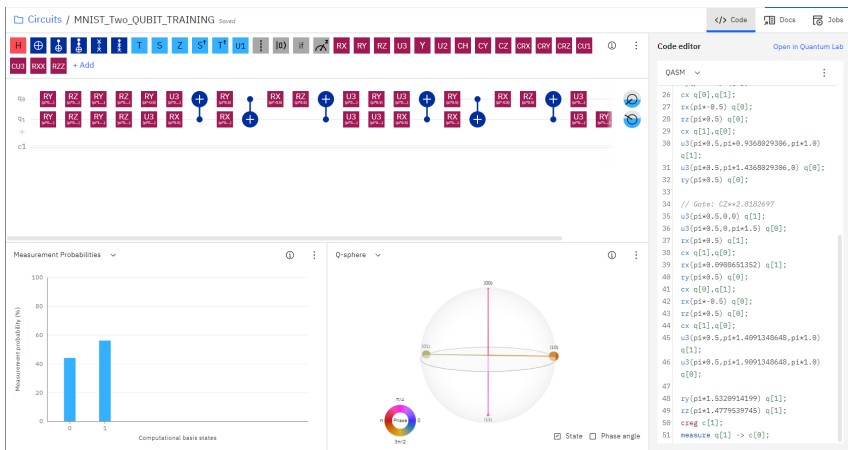

Figure 12: Quantum circuit for binary classification on MNIST dataset (T2)

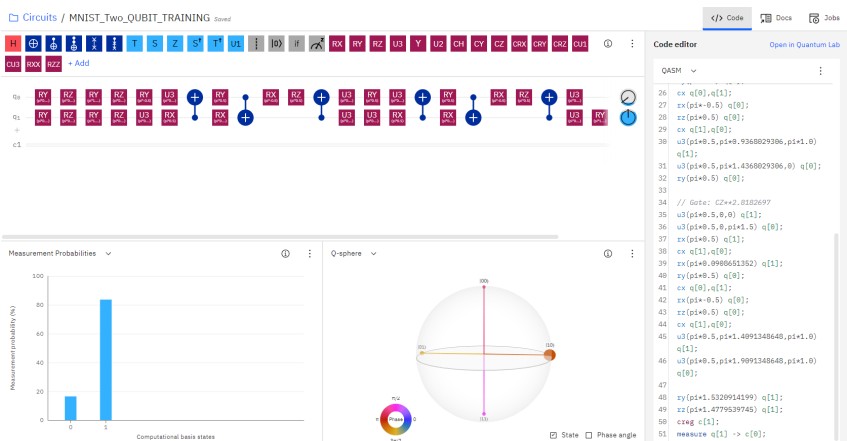

Figure 13: Quantum circuit for binary classification on MNIST dataset (T3)

