# OpenReview forum: "GenQu: A Hybrid System for Learning Classical Data in Quantum States"
_ICLR.cc/2021/Conference — Reject_

### Official Review · AnonReviewer4 · 2020-10-27
**The paper is well written, very easy to follow.  But the lacking part is its limited technical contributions.**

**Rating:** 3
**Confidence:** 5

**Review:**

This paper presents GenQu, a hybrid and general-purpose quantum framework for learning classical data through quantum states. By encoding two dimensions of data per one qubit. they demonstrate the effectiveness of their framework via two classical classification tasks, where 1 and 2 qubits are used, respectively. This paper is more like an entry-level tutorial, rather than a technical paper.  More technical contributions are needed towards a paper.


1. One of the key contributions claimed by the author is that "they show the power of encoding two dimensions in one qubit ...This thereby reduces the quantum state dimensionality by 2^(n/2)." I am super surprised by this claim. What is your baseline? One qubit per one dimension of classical information? Could you refer to the paper from which you get this baseline? There are just too many papers[1] about how to encode classical data into quantum states.  The coding scheme proposed in the paper is not novel and not even state-of-the-art.

2. What is the key difference between your framework and TensorFlow Quantum[2]? For me, TensorFlow Quantum is a much stronger framework. For example, the SINGLE QUBIT KERNELIZED CLASSIFICATION case study in section 3.3 is just an illustrative example in [2].

[1] Biamonte, Jacob, et al. "Quantum machine learning." Nature 549.7671 (2017): 195-202.

[2] Broughton, Michael, et al. "Tensorflow quantum: A software framework for quantum machine learning." arXiv preprint arXiv:2003.02989 (2020).

---

> ### Author Response · Authors · 2020-11-19
> **Reply to Reviewer-4**
>
> We thank the reviewer for your comments. To address the concerns,
>
> - This is not an accurate claim. We have removed it in revision. Our paper's main contribution is that the system is built on top of the real quantum computers. We use the three representative examples to demonstrate the effectiveness of our system.
>
> - The key difference between the proposed system and Tensorflow Quantum (TFQ) is that of a simulator, and our proposed system works with a publicly available quantum platform. In the revision, we compared the MNIST example with Tensorflow Quantum though the comparison can only be performed in the simulator. While many different models have been proposed in the literature, most of them are based on simulators or specifically designed hardware, which are not publicly available.

---

### Official Review · AnonReviewer2 · 2020-10-27
**Review of "GenQu: A Hybrid Framework for Learning Classical Data in Quantum States"**

**Rating:** 3
**Confidence:** 4

**Review:**

I agree that promoting experiments for real quantum hardware is important. But I don't think the team has yet to create a working platform that could be accepted to ICLR. If the open-sourced platform is the main contribution (rather than a new understanding of how quantum computers could be useful for machine learning problems), then the authors should submit the manuscript after having the open-sourced software available. Furthermore, a lot of the wording should be changed (the current version sounds like they are proposing a new quantum machine learning framework, while they are creating an open-sourced platform).

##########################################################################

Summary:

The authors propose a framework, GenQu, for learning classical data using quantum computation. The classical computer would encode the classical data into quantum circuits. The quantum computer would then run the quantum circuit, measure the resulting quantum state, and feed the measurement data back to the classical machine. This process would repeat until the classical computer output the final result.

##########################################################################

Reasons for score:

This framework is not new and has been widely adopted in the quantum machine learning community. It is unclear to me what is being proposed by this work. The framework is known as a variational quantum-classical algorithm and there is extensive literature for different applications in quantum computing, such as quantum chemistry, simulating quantum field theory, optimization, and machine learning. For example see references [1, 2] for existing proposals for machine learning applications. Due to the lack of meaningful contributions, I would not recommend acceptance.

##########################################################################Pros:

Cons:

1. The framework is not new. It is not scientifically correct to claim the proposal of a new framework "GenQu" when this has already been widely adopted in the quantum machine learning community.

2.  The authors did not provide any new theoretical insights into how quantum computation can learn classical data better.

3. The numerical experiments were not strong enough to justify any form of advantage using the quantum computer. Furthermore, these numerical experiments have already been presented in the literature. For example, a tutorial in Tensorflow Quantum [3] has also included such an experiment.

#########################################################################


[1] Havlíček, Vojtěch, et al. "Supervised learning with quantum-enhanced feature spaces." Nature 567.7747 (2019): 209-212.

[2] Farhi, Edward, and Hartmut Neven. "Classification with quantum neural networks on near term processors." arXiv preprint arXiv:1802.06002 (2018).

[3] Peruzzo, Alberto, et al. "A variational eigenvalue solver on a photonic quantum processor." Nature communications 5 (2014): 4213.

[4] https://www.tensorflow.org/quantum/tutorials/mnist

---

> ### Author Response · Authors · 2020-11-19
> **Reply to Reviewer-2**
>
> We thank the reviewer for your comments.
>
> - The proposed system is based on a publicly available quantum platform, not simulators or special hardware. Therefore, we think it is not comparable to the literature listed [1-4]. Reference [1] has the full privilege on the quantum machines, not the public version. Reference [2] only uses classical simulations, as the authors stated, "Our work is exploratory and relies on the classical simulation of small quantum systems." Reference [3] utilizes a specially designed quantum processor on top of a classical computer, clearly not available to the community.
>
> - Reference [4] Tensorflow Quantum (TFQ) is a simulator that can not directly run on quantum machines. We have designed APIs to convert Tensorflow Quantum code and execute it on IBM-Q with our proposed system. While the comparison can only be done in simulation (TFQ is a simulator), we have compared the method in [4] in the revision. The result shows that we can significantly improve the accuracy. This is due to the encoding methods, where we use exceptions, and [4] utilize binaries. As the proposed system's main contribution is that it builds on top of a publicly available quantum computing platform, it is a feature that our system can accommodate different algorithms easily, which makes the system more practical. When open-sourced, we plan to make it support multiple real quantum computing platforms, e.g. IBM-Q and Amazon Braket.

---

### Official Review · AnonReviewer3 · 2020-10-28
**GenQu: A Hybrid Framework for Learning Classical Data in Quantum States**

**Rating:** 2
**Confidence:** 5

**Review:**

The paper claims to introduce a new quantum machine learning framework called GenQu. However, the description of the framework very vague (using classical computers to optimize the parameters of a fixed quantum circuit), and hardly novel. In fact, the same basic ideas are so well-known in the community that they are described in detail as usage examples for popular quantum computing platforms such as Qiskit and IBM Q.
The only remotely nontrivial part of the paper is contained in Section 4.2 about Quantum Deep Learning, where the authors consider the MNIST data set. Upon closer inspection it turns out that they use PCA to reduce the dataset to 4 dimensions, which is in turn used to train a "quantum neural network" to perform binary classification (i.e. to discriminate between '0'-instances and '5'-instances). The authors claim that such a quantum classifier provides an advantage versus a convolutional neural network in terms of
 1. the number of training epochs (while ignoring the time needed to perform PCA), and
 2. the number of parameters (while ignoring the parameters needed to describe the principal components).
Additionally, no confidence intervals are visible on Fig. 7, which suggests that the data might have been obtained from a single experimental run. Finally, there are several instances of sloppy writing, such as the inconsistent usage of math mode for variables, the statement P(|\phi>) = |0>, the typo "iWs" instead of is, etc.

---

> ### Author Response · Authors · 2020-11-19
> **Reply to Reviewer-3**
>
> We thank the reviewer for your comments.
>
> Our main contribution in the paper is that the proposed system is based on a publicly available Quantum Computing hub. While it is easy to develop a simulator-based solution, due to limited access to real quantum machines and the fair sharing policy on IBM-Q, it is challenging to train a real hybrid system model.
>
> The IBM-Q fair sharing policy introduces overhead for machine learning algorithms and other intense quantum applications requiring multiple communication between the user and the quantum algorithm. For example, running simple ML classification on the quantum computer requires multiple tunning and resubmission of circuits. In this case, fair share always puts the new generated circuit on the end of the IBM-Q machine, which causes an unpredictable delay to train or complete the quantum algorithm. We present the 3 representative examples to illustrate the effectiveness of using the proposed system to train a real quantum computer model instead of using simulators.
>
> To answer the two items,
>
> 1. We feed PCA data to both CNNs and QNNs. Therefore, we think it is a fair comparison. When considering the publicly available quantum computers with 5 to 7 qubits, it is simply not possible to train a full-size MNIST dataset.
> 2. IBM-Q quantum computers are in different settings and locations, and due to the random noises on different computers, we did not provide confidence intervals.

---

### Official Review · AnonReviewer1 · 2020-10-29
**This is one of the early papers on the concept of using quantum qubits for neural network computations.  The authors propose a robust method to map data into entangled qubits; this method is efficient in the qubit space, which is an important consideration in quantum systems.  However, the actual optimization problem is not solved by the quantum computer, only the forward pass.  I see this as a start in a potentially new and important field.**

**Rating:** 4
**Confidence:** 4

**Review:**

The authors identify that “classical” learning is running into limitations due to power and scale of computing systems. The authors suggest “quantum” learning might supplant classical learning and solve these fundamental challenges.  And, the authors suggest a method that is efficient in the number of QuBits, which can be quite precious.
The authors believe this work is well motivated by the fact that little work has been done in finding ways to combine the virtues of quantum methods with the power of deep neural networks.
The authors propose an encoding scheme to map data efficiently to a qubit, to exploit multiple qubit samples. The ability to entangle multiple qubits to represent points on the Bloch Sphere is a key concept for efficiently using the qubit space.  I believe the biggest contribution this paper makes is this efficient mapping of data to qubit space.

Overall, this paper demonstrates that some ASPECTS of neural computing can be mapped to a quantum, qubit system.  This is a timely topic as the scale and energy usage of classical computers for deep learning training is becoming untenable, and many are thinking about ways to make this problem tractable.  The most novel concept in the paper is the symmetry described between the nature of how entangled qubits can encode information, and how a neural network progressively makes data more separatable.  The actual learning process appears to be done on a classical computer (backprop and update) and then mapped back into the quantum system.  To me, a much more interesting result would be exploiting the quantum nature of the computer to solve this non-convex optimization problem of reducing the loss via layers of features. The generative aspect described is a nice side-effect (as the authors indicate) but isn’t really a novel concept.
The results of the paper really are proof that the basic functionality is there, but not really a proof of concept in a general way.  The dimensionality of a small neural network for MNIST had to be reduced to be able to be realized on an actual quantum system; this weakens the claims of these results as it may be untenable as complexity is scaled-up.  Nevertheless, I believe this result does demonstrate an existence proof, and the community will benefit.

In general, the language should be cleaned up.  There are just a few sentences with a slightly odd structure.

Section 3: I can see how the data is encoded to make it linearly separable.  But how does this relate to backprop? I can see that backdrop isn't really handled by the quantum system, so the separable mapping of the data is really just a general data concept.

Section 3.1: It looks like the actual backprop is done on a classical computer.  A much more interesting result would be to do the actual gradient descent in the quantum domain.

Section 3.2: Is the data encoding done on a classical computer?

Section 3.5: Looks like the generative concept is really just exploiting the stochastic nature of qubits as a random generator.  I think this points to a strong parallel between neural networks and quantum systems but isn’t really a new concept.

---

> ### Author Response · Authors · 2020-11-19
> **Reply to Reviewer-1**
>
> We thank the reviewer for the appreciation and comments.
>
> This paper's main contribution is that the proposed system is built on top of a publicly available quantum computing platform. We use the 3 representative examples to demonstrate the effectiveness of the system. When open-sourced, in addition to simulations, we hope the proposed system could encourage the community to conduct experiments on real quantum computers (e.g. IBM-Q and Amazon Braket).
>
> To answer the concerns,
>
> - Yes, the parameter tuning is done on classical computers. The quantum circuit needs to be updated when the parameters changed. However, due to the low qubit count available on the current quantum machines (5-7 qubits), it is not feasible to update the circuits dynamically as it needs to double the required qubits of a given model.
> - Data encoding is done on the classical computer as access to the IBM-Q quantum machine is limited.
> We added a PCA MNIST example in the generative nature section.

---

> > ### Comment · AnonReviewer1 · 2020-11-21
> > **right journal?**
> >
> > I was generous with my score as I am open to a diversity of topics to be included.  However, looking at the rest of the reviews, I think you might want to reconsider which journal/conference this work is included.  Solving a NN problem using the fundamental characteristics of these machines is the most interesting result, but this paper doesn't really show that.  I don't see this as solving a real problem using a quantum machine; but rather, adapting a quantum machine to do something that classical machines can also do.

---

> > > ### Comment · AnonReviewer1 · 2020-11-23
> > > **score updated**
> > >
> > > I have updated my score based on relevance to this journal/conference.  Thank you for submitting interesting work!

---

### Decision · Program_Chairs · 2021-01-07
**Final Decision**

**Decision:**

Reject

**Comment:**

This paper proposes a new quantum machine learning framework which is evaluated on the MNIST dataset. While the paper was relatively well written, reviewers noted that most of the ideas are already well established and used in quantum machine learning community. Thus it was not clear what novelty is provided relative to related work.